# Navigating life when a loved one's (euthanasia) death is near: A narrative interview study from the Netherlands

Bernadette Roest[1], Megan Milota[2], Carlo Leget[1]*

1 University of Humanistic Studies, Utrecht, The Netherlands, 2 Julius Center for Health Sciences and Primary Care, University Medical Center Utrecht, Utrecht, The Netherlands

* c.leget@uvh.nl

## Abstract

In this article, we describe our empirical research that started out as an exploration of "family involvement" in the Dutch practice of euthanasia, in the broader context of end-of-life decision-making and care under guidance of GPs in the home-setting. Informed by care-ethical insights and narrative approaches to qualitative research, we performed an in-depth interview study with 18 close relatives of people with incurable metastasized cancer (9 prospective, 9 retrospective). We came to understand how relatives' involvement—not only in euthanasia but in any mode of dying—cannot be separated from relatives' efforts to navigate through many dimensions of life when the death of loved one is near. Relatives have to navigate different spaces, decision-dynamics and the unfolding of professional care, strong emotional experiences, and intimate relationships ("the I-you-we"). This study brought to the fore that relatives and patients first and foremost accompany each other on this journey. The role of the GP can be valuable but vulnerable, and relatives' broader social network and other professionals can be of enormous importance. The results of this study invited us to shift our perspective: it is not family members who get involved in euthanasia as a primarily medical affair. Instead, medical professionals are taking part in the profoundly social-relational experience of death and dying within families, whether that entails euthanasia or not. With this shift of perspective, specific practical and ethical questions start to receive more attention, for example questions about the available support for both patients and relatives regardless of the mode of dying.

## Introduction

Family involvement in healthcare in general, and end-of-life care in particular, has become a topic of great interest in sociology, bioethics and palliative care in the last decennium. Scholars have, for example, drawn attention to patients' preferences regarding family involvement in decision-making [1]; the practical challenges of

**Data availability statement:** Data cannot be shared publicly because of the privacy of the research subjects on this very sensitive topic. Data are stored at a secured disk at the University of Humanistic Studies and are available on request, for researchers who meet the criteria for access to confidential data, contacting the Institutional Data Access: datamanagement@uvh.nl

**Funding:** The author(s) received no specific funding for this work.

**Competing interests:** The authors have declared that no competing interests exist.

researching and delivering family-centered palliative care [2,3]; and concepts such as relational autonomy [4] and an "ethics of families" [5,6].

Assisted dying forms a peculiar "case" in this interdisciplinary academic study into family involvement in decision-making and care. Empirical research from various case sites has demonstrated involvement of family and friends on many levels of practices of assisted dying [7,8]. Meanwhile, the moral and legal justification for assisted dying often relies on the choice and suffering of the individual patient, for example in the Netherlands where legislation on assisted dying has existed since 1994 [9]. As a consequence, ethical issues that have received a lot of attention relate to matters of "undue influence" of relatives during decision-making, i.e., pressure, conflict or disagreement [10,11]. Another issue that has received attention is the complex relationship between patients' autonomous choices, feelings of being a burden, and a "duty to die" that might come with legislation of assisted dying [12,13].

In the Netherlands, assisted dying is most often performed in the form of euthanasia, i.e., the administration of a lethal substance through an intravenous canula by a physician, on explicit request of the patient. Euthanasia accounts for 97.4% of all cases of assisted dying [14]. General practitioners (GPs) are still the physicians who perform euthanasia most often, namely 80.4% of all registered cases of assisted dying in 2022. In addition, euthanasia most frequently occurs at home (79.6% of cases) and with incurable cancer as the underlying illness (57.8% of cases) [14].

The Dutch GPs are traditionally seen as the ones closing the gap between individuals and families in the practice of euthanasia. They are supposed to have a long-lasting treatment relationship with both patients and relatives and are primarily responsible for palliative care at home [15–17]. Thereby, they would be in the ideal position to assess suffering and voluntariness related to euthanasia requests and concomitantly keep an eye on relatives during and after euthanasia.

However, while Dutch GPs' experiences with euthanasia have been widely studied—including the difficulties they may encounter with relatives—the experiences and needs of family members themselves have received far less attention [7,8]. In addition, some important dimensions of family involvement in the practice of euthanasia remain understudied as they are perceived to be disconnected from palliative care, or because studies focus solely on the legislative framework [18]. Meanwhile, signals are emerging about diverging expectations between relatives and patients and their GPs about what they might expect of euthanasia and other care at the end of life [19,20].

Therefore, we decided to further explore family involvement in the Dutch practice of euthanasia in the broader context of end-of-life care at home under guidance of GPs, by performing a narrative interview study. In this article, we describe the results of our study and the practical and conceptual questions that have arisen. In addition, we describe the research-process in such a way that readers can follow our own process of changing our perspective on "family involvement in euthanasia". Thereby, we hope to offer new insights to scholars from both in- and outside the Netherlands who are interested in practical and theoretical dimensions of end-of-life care and the possible place of assisted dying therein.

## Methods

The moral epistemology of MU Walker formed the foundation of the empirical study described in this article [21]. Walker argues there is a thorough intertwinement of the moral and social dimensions of life. People holding different social positions may not share the same understandings about what matters most morally in specific situations in daily life; empirical research can help unravel what is most at stake for the different people involved [18,21]. In addition, the study was informed by the broad field of care-ethics, in which needs and responsibilities are seen as important guiding concepts in empirical research [18,22]. Furthermore, our empirical study should be placed on the interpretive pole of the spectrum of scientific paradigms that can be discerned in empirical-ethical research [23]. Specifically, we adopted a narrative approach to qualitative interviewing and analysis that slightly differs from commonly used approaches such as thematic or content analysis [18,23]. We describe the specific methodological steps in the paragraphs below.

The study was conducted as part of a larger research-project (50 interviews in total) exploring the needs and responsibilities of patients, GPs and other professionals during end-of-life decision-making and care, including euthanasia, in Dutch general practice. We included both a retrospective and a prospective group in this sub-study. We recruited 1) bereaved relatives of patients with cancer who had received euthanasia by the GP at home or had seriously considered it, and 2) relatives of patients with incurable, metastasized cancer who were willing to speak about their ideas with regard to future end-of-life decision-making and care for their ill relative under guidance of the GP, including the possibility of euthanasia. We chose this design in order to capture experiences with euthanasia decision-making in the broadest sense, and not just the phase in which patients pose an explicit request for euthanasia to their attending GP. This "formal" part of euthanasia decision-making has already been explored in previous qualitative research [24].

Recruitment of participants entailed a process of network- and snowball-sampling, in which we tried to obtain as much variety as possible in the sample on the domains of age, gender, kind of relation to the patient and type of underlying malignancy. In addition, we aimed at sufficient "information power" in line with our narrative approach and explorative study aim [25]. For the retrospective group, possible participants were contacted via GPs in the extended professional networks of the authors, and via mailings send out by contact-persons in the Dutch Federation of Cancer Patient Organizations and an online platform for cancer patients. For the prospective group, patients with cancer who were initially contacted for the other sub-study through the same means were asked if they had a relative who wanted to participate as well.

Eligible participants first received a short letter with information about the study and the contact details of the principal investigator (BR) and were asked to contact her themselves in order to express interest in participation. After a conversation about the study via telephone and reading the full information leaflet, participants were scheduled for an interview and informed consent forms were signed. An interview was scheduled with 18 participants (9 in each group). Participants were interviewed in their home (n = 8) and later via video-call (n = 10) due to COVID-19 related restrictions, between November 1, 2019 and October 1, 2020. The interview-duration was between 27 and 98 minutes (mean 59 min.).

BR conducted all interviews. She is a GP and bioethicist with additional training in ethnography and narrative-oriented qualitative research, as well as close-listening and close-reading techniques as taught in Columbia University's Narrative Medicine Program [26]. All participants were aware that BR was a GP before the start of the interviews, but there was no previous treatment or working relationship with the participants, their ill relative or their attending GP.

BR took a narrative approach in interviewing, starting with a few open questions and followed by explorative follow-up and probing questions, allowing the participants to speak in detail about what mattered most to them in relation to the illness and (future) death of their ill relative. The interviewees were first asked about themselves ("What do I need to know from you so that I can understand you well?"), followed by the prompt "Please tell me about your relative and his/her illness".

If the interviewees did not speak about it spontaneously, the interviewer asked about the relationship with the GP, sources of support, and felt responsibilities and needs. When interviewees found it difficult to name their needs, the interviewer reposed the question as "what would your advice be to others who come in the situation you are/have been

in?". BR ended each interview by asking the participant if there were things that weren't discussed yet, but which were of importance to them.

From the information leaflet and introductory conversation with BR, the participants knew that euthanasia could be one of the topics in the interview. During the interviews, BR followed the participants' lead in addressing the topic. As soon as they brought up euthanasia, BR invited them to expand on the topic. If participants in the prospective group did not spontaneously bring up euthanasia, BR carefully probed for (future) end-of-life choices of their loved ones in more general terms, following the participants' language. If this exploration evoked hesitance or strong emotions on the participant's side, BR paused the interview to decide with the interviewee about whether or not to continue, or to switch to another subject. In this way, we tried to balance methodological and research-ethical considerations, i.e., obtaining rich data while not interfering too much with the participants' own thought-process or the phase of decision-making they might be in with their ill relative and attending healthcare professionals.

Fieldnotes were made after each interview and two weeks later a follow-up telephone call was scheduled to give participants the opportunity to express additional thoughts or questions and reflect on the experience of participating in study. Fieldnotes were made of these moments as well. In addition, BR offered to send participants a narrative summary of their interview. The participants who agreed to this (n = 17) were given the opportunity to contact BR again by e-mail of phone after receiving the summary. In total, ten participants responded to the narrative summaries via e-mail or telephone (n = 7 resp. n = 3).

All interviews were transcribed, anonymized, and then read and discussed by BR and CL. Afterwards, BR performed an analysis of each individual transcript in different cycles: first, a cycle of inductive coding of emerging themes conform narrative thematic analysis, a second and third cycle of coding through the lenses of contextual levels and narrative features (person, place, time, mood, metaphor etc.), and fourth a wrap-up of what one specific interview contributed to answering the research question. CL and MM analyzed a subset of interviews via this approach; differences and similarities in their analyses were also discussed. For examples of this analytical process, we refer to our previous publication [23].

After this detailed analysis of the individual interviews, we performed a cross-case analysis to come to meta-level understandings of the entire set of narrative interviews. BR compared the analyses of the individual interviews in close consultation with the other members of the research team. She sought to identify those parts of the analyses that emerged as most prominent and most informative in light of the research question, earlier empirical research, and the day-to-day clinical and academic discourse on euthanasia and end-of-life care. BR formulated descriptions that could capture these findings without losing the ambiguities in and contrasts between the individual interviews. This step was again discussed among the members of the research-team and the descriptions were further fine-tuned. To help the reader understand how our results contrast with earlier empirical research and common understandings about the Dutch practice of euthanasia and "family involvement", we briefly refer to these elements at the start of each subsection in the result-section.

For the representation of results, BR and MM wrote narrative vignettes displaying summaries of a part of the individual interviews including quotes. In this way, we were able to achieve a thick description of results without losing the interrelatedness of people, places, actions and considerations typical of narrative. Although details had to be altered to prevent recognizability, we carefully chose how to change these details in order not to steer too far away from the original interviews. The gender of the interviewees, their ill relative and the attending GPs was not altered since it is possibly a parameter of importance, in need of further inquiry. For other characteristics, such as occupation and diagnoses, similar alternatives were sought. All the names in the vignettes are pseudonyms.

For the sake of brevity and readability of this article, some of the vignettes have been placed in a supplementary file. Still, the reader will get an impression of all vignettes and the variety of experiences they convey through short summaries placed in the main article. In addition, the results-section contains several quotes from interviews that were not displayed in the vignettes to add even more depth to the description of the findings.

 

For member check-purposes, two people independent from the research team, representing relatives' voices in healthcare and/or having experiences as a bereaved relative themselves, received a first version of this article. They were asked to provide feedback on how the study results resonated with their own experiences or with the experiences of the people they represented.

Reflexivity was seen as an integral part of the whole research process and it was operationalized through continuous discussions in the research team about preliminary (moral) positions and personal experiences with illness and dying, keeping both a methodological and reflective diary, and frequent debriefing by the first author with fellow-researchers and with a pastoral counselor independent from the research-team.

The research question that guided this sub-study was: what can we learn from co-constructed interview narratives about the needs and felt responsibilities of cancer patients' relatives during euthanasia decision-making in the broader context of end-of-life care in the home-setting? In particular, we were interested in how relatives' needs and responsibilities take shape in interaction with their GP.

### Research ethics

The Medical Ethical Review Committee of the University Medical Center Utrecht reviewed the study protocol (protocol 19–303/C) and judged that it did not meet the criteria listed under the Dutch Medical Research Involving Human Subjects Act (WMO), therefore a waiver for further evaluation by the Committee was granted. Consultation with ethicists, social scientists and healthcare professionals with expert knowledge on the topic of (studying) assisted dying took place before the onset of the study and on few occasions during the study when new research-ethical questions emerged.Participants were extensively briefed about study aims and procedures, including the possibility to withdraw from the study or stop the interview if it became too burdensome. Written consent was obtained from all participants. In case of video-call, their consent was recorded on audio. Participants were told they could contact BR afterwards in case of remaining questions or a wish to receive aftercare. In addition, they received contact details of an independent consultant who they could contact in the case of questions or complaints they could not discuss with the researchers. Follow-up contact took place with all participants (see methods). Participants described their experiences of participating in the research as positive, even if it had been emotional to recount their experiences. Aftercare—in the form of an additional visit, phone call or referral to a healthcare professional—was provided to three participants.

### The Dutch context and definitions

The Dutch legislation on assisted dying is still part of penal law. In addition, it is still considered an extraordinary medical procedure according to the Royal Dutch College of Physicians. Assisted dying currently accounts for 5.1% of annual deaths in the Netherlands [14]. However, euthanasia is "out in the open" via screenings of euthanasia-deaths in documentaries broadcasted on public television, but also through provision of information about euthanasia as one of several choices at the end of life on government-issued websites [27].

In the remaining part of this article, we use euthanasia and assisted dying interchangeably. In the Netherlands, the vast majority of assisted dying-cases consists of euthanasia and not physician-assisted suicide (i.e., the provision of a lethal drink by the patient which is then ingested by the patient him/herself, only 2.1% of cases) [14]. In our study sample, we only came across cases of euthanasia and not of assisted suicide. In addition, we use palliative sedation as an equivalent for continuous deep sedation until death, conform regular use of these terms in the Netherlands.

### Results

The participating relatives were spouses (n=13), offspring (n=4) and a sibling (n=1) with ages ranging between 25 and 76 (mean 54 years) and an equal division of men and women. All participants were white and they described their life-view as non-religious/agnostic/atheist (n=11), protestant or catholic (n=3), humanist (n=2), or Christian non-practicing (n=2). The primary diagnoses of

the ill relatives included gastro-intestinal cancers, lung-, breast- and urogenital-cancer, neuroendocrine, skin and soft tissue tumors. In the retrospective group, the time between the death of the patient and the interview was 2 till 36 months (mean 13 months). The age of the deceased patients ranged from 40 till 74 years (mean 63). In seven out of nine retrospective cases, the interviewee's relative had died by euthanasia, in the others it had been seriously considered. In the prospective group, the ill relative of two interviewees died within eight months after the interview took place. BR had follow-up contact via telephone with one of them and the corresponding fieldnotes were included in the study. The other participant did not respond to further requests for contact. Unless otherwise stated in the subsequent sections, the results presented pertain to both the retrospective and prospective groups.

This narrative interview study showed us that relatives' involvement in euthanasia cannot be seen apart from their efforts to navigate many more dimensions of life when the (assisted) death of a loved-one is near. And we came to realize that "family involvement in euthanasia", the notion we started with in the first place, already presupposed a medical or professional's gaze. The following results invited us to shift our perspective: it is not family members who get involved in euthanasia and other modes of dying as primarily a medical affair. Instead, we may consider the perspective that it is GPs and other professionals who get involved in death and dying as a family affair.

**Navigating life when a loved one's (assisted) death is near**

The cross-case analysis made it clear that we could not understand relatives' needs and felt responsibilities related to both euthanasia and other end-of-life decisions and care separately from their multilayered stories addressing many other dimensions of life. In other words, relatives' *lives as a whole* were at stake, not just their needs and responsibilities, or any specific "roles" in decision-making. In addition, while we had sought to understand what happened at home in interaction with GPs, interviewees spoke at length about other places and other people who were immensely important when considering care and choices at the end-of-life including euthanasia. Nicole's vignette (1) provides a first example of such a layered life, layered losses, choices and care transcending the home, and how an euthanasia-death may fit in.

What all interviews had in common, is that the relatives had spoken extensively about how they *had to find their way* through different dimensions of life while their loved one's (assisted) death was near, a process best captured by the metaphor of *navigating.* To be clear, we associate this metaphor with a searching attitude, finding one's way through uncharted territory, with limited means and multiple uncertainties about the final destination. We do thus not adopt a more technical interpretation of "navigating", as moving along clear pathways with predefined milestones and endpoints.

We summarized the different dimensions of life that relatives have to navigate as *different spaces, decision dynamics and the unfolding of professional care, strong emotional experiences and intimate relations (the 'I-you-we').* These four dimensions reflect and summarize those narrative elements (place/time, actions, emotions, people) that emerged as most prominent from the individual interviews and cross-case analysis in comparison to earlier empirical studies on the topic of family involvement in assisted dying [7,8]. The metaphor of navigating also helped us to be attentive to the question of *who navigates with* the relatives: the navigating seemed first and foremost done among and by families and their inner circle themselves. Some found great support from their GP or other professionals, while in other cases such support was missing.

In the following section, we flesh out these different life dimensions that relatives have to navigate when the (euthanasia) death of a loved one is near. In the presentation of results, we alternate descriptions of the different dimensions with vignettes in which these dimensions come to the fore at the same time, albeit with varying emphases. Thereby, we illustrate how relatives' stories can be disentangled for analytical purposes but cannot actually be neatly divided into separate sub themes. We have included one vignette (Nicole) in the text; the other vignettes can be found in S1 File.

**Vignette 1: An example of layered lives, layered losses**

Nicole (45 years, former employee in a commercial enterprise) was about to lose her husband, her job, and her home as a place of safety at the same time. Now that everything has passed, she struggles with the aftermath of it all. Her husband was diagnosed with a widely metastasized melanoma after their GP first mistook his complaints for something

innocent. She was devasted to learn about the diagnosis. "Then your world turns upside down." Nevertheless, she tried to keep their life going amidst her husband's palliative systemic treatment and frequent interventions when complications occurred. Meanwhile, Nicole tried to keep up with responsibilities at work, but the compassion for her situation and her frequent absences started to wane.

Several times, Nicole's husband ended up in the hospital and even the ICU due to life-threatening complications of his cancer. At one point her husband became unconscious, and Nicole was asked to decide about treatment (dis)continuation. She opted for continuation and he survived. Although she was at peace with her decision in the end, the memory kept returning. "Maybe I should have let him go then and saved him all the later suffering."

When her husband's condition further worsened and treatments in the hospital stopped, they opted for homecare. "We both wanted care at home, no matter what happened, so he could die at home." But she experienced the arranged home- and night care as "horrible," "so unrestful." Carers came at irregular times, intruded into private spaces, or lingered in the house longer than she wanted. She did not dare to speak up to them, "maybe a bit out of fear that while I may not need them today, I might urgently need them tomorrow." Nevertheless, she felt they had lost their home as a place of rest. They found a solution by moving her husband to a hospice, while letting him come home during some weekdays. She continued to do the daily care herself. "I always wanted to care for him myself. I enjoyed doing so and he was happy with it too."

With their GP—a new one as they lost trust in the other because of the misdiagnosis—they started speaking about euthanasia. "For us it was clear that we did not want…like…extreme suffering. But in the end you push your boundaries and…in hindsight it was a lot of suffering" "We were 80% certain it would end that way [i.e., euthanasia]." "If you are in such a situation, that you are so ill, then you would be happy if you can be assisted, that it ends under your terms, so to say." They were grateful for the clarity in conversations and speed with which their new GP acted, also when her husband's condition rapidly began to decline, and they wanted to let the euthanasia take place at their home earlier than scheduled.

Nicole says that she was extremely sad but relieved to say goodbye in relative rest. Her husband finally died at home through euthanasia after having said goodbye to his parents and siblings. She found some consolation in the fact that her husband had started to dream of deceased family members in the days before his death. Meanwhile, the situation at work had escalated and a reintegration trajectory had not worked out. Not long after the death and burial of her husband, she lost her job.

While the GP has contacted her and provided some aftercare, Nicole still searches for more specialized grief-counselling and some closure for everything that happened in the hospital. She says that earlier bits of information regarding practical possibilities at the end-of-life and the course of dying would have been helpful. "In the last phase, you have absolutely no idea what to expect, and that makes it scary." She admits that she and her husband were not inclined to talk about those things earlier in the trajectory because "it's frightening."

## Navigating different spaces: hospitals, homes, workplaces

What first stood out in the relatives' interviews was the multitude of physical spaces they and their ill relatives had to navigate while dealing with the illness and impending (assisted) death of their loved ones, all involving different people and separate responsibilities. We summarized them as *hospitals, homes and workplaces*. We had originally intended to focus on the home-setting only, since it is the place where euthanasia takes place most often and which is traditionally spoken about as the ideal place to die in the Netherlands. However, the interviews pointed at how additional spaces had an enormous impact on relatives' and patients' perceived burdens and received care, and thus on future decisions about for example euthanasia. See the vignettes of Christopher (2) and Katherine (3) for an illustration of these elements. Furthermore, some interviews challenged the ideal of the home-death: "the home" emerged as a space of continuing (family) life which could be severely disrupted by the presence of professional carers.

Many interviewees first spoke about accompanying their ill relatives during multiple in-hospital treatment regimes, regularly supervised by several specialists and/or nurses and sometimes even spread over different hospitals. *Hospitals* could be the primary place of treatment until a couple of months, weeks or even single days before the (euthanasia) death, see the vignette of Ryan (son, vignette 7) for an example of the latter. While some interviewees spoke about finding good care and recognition for both their ill relative and themselves, others spoke about not finding access, not being taken seriously, or suddenly being designated as primary decision-maker about matters of life and death because of a patient's sudden unconsciousness.

With regard to *homes*, many interviewees talked about extensive caring-responsibilities for their ill relatives, ranging from daily physical care, wound and stoma care, to rebuilding houses and overseeing finances in the wake of the illness of their loved ones. These responsibilities could take from weeks up to years, with intensity changing over time. Homes were also spoken about as places of rest, or places filled with memories of previous family gatherings and peaceful deathbeds of beloved others (vignette 3, Katherine). In addition, the home was also spoken about as the place of social and family life that could change tremendously due to illness and impending death, for example due to emergent family disputes or the presence of home care services. The vignette of Nicole (1) provides an illustrative example of the possible disruptive effects of home care services. The vignette of Christina (11) illustrates how the impact of caring for a loved one at home could be huge even without the presence of home care services, and regardless of the mode of dying. In the case of Christina, she had anticipated and accepted this.

Few interviewees also spoke about finding their way—often unexpectedly and temporarily—through hospices and care-homes, as a substitute for the own home when care-demands exceeded what they could deliver. Furthermore, it was remarkable to see how several interviewees had to deal with acute and chronic health issues of their own while caring for their ill relatives, see for example the vignettes of Christopher (2) and Sophia (4).

Last, interviewees frequently and spontaneously referred to navigating *workplaces*: several mentioned worries or actual problems concerning job and financial insecurity in relation to the illness and care responsibilities of their loved one (vignettes 1, Nicole, and 10, Charlotte). Other interviewees spoke about the opposite: how they found support and a welcome distraction in paid or voluntary work. In addition, some interviewees mentioned how they felt free to focus on the care for their relative because they were retired (e.g., vignette 5, Peter) or were business owner (vignette 11, Marc) and could easily schedule their activities themselves.

### Navigating decision-dynamics & the unfolding of professional care

A second aspect that emerged in this study was that interviewees had to *navigate decisions* together with their ill relative. However, this encompassed much more than just the question of whether or not to pursue euthanasia, and entailed more than several conversations with one GP. Instead, the interviewees spoke of multiple decisions over time, with multiple considerations involved. In addition, many interviewees mentioned how earlier priorities and decisions could change, often in response to care that did (not) unfold in interaction with a variety of professionals or different GPs. We therefore found that the commonly used notion of "decision-making" did not fully capture the relatives' experiences on this point; the process seemed much more dynamic and more context-dependent than previously described [7,8]. This led us to the overarching description of *"navigating decision-dynamics and the unfolding of professional care"*.

First, several interviewees in both the prospective and retrospective group mentioned how they themselves already had been put in the position to make far-reaching decisions for their ill loved one before any GP-led conversations about the end of life had taken place, due to life threatening complications of the illness or treatment. See the vignette of Nicole (1) for an example.

In addition, many interviewees expressed a wish for their ill relatives to have some control or say about their end of life. Interviewees also frequently mentioned the wish for their loved one to die at home and to avoid suffering, as well as sufficient rest and time to say goodbye and avoid stressful situations at the end of life. However, these things were not always

achievable. Several interviewees recounted that while "shared decisions" or "plans" had already been made in consultation with a GP, events unfolded differently due to the (im)possibilities of the healthcare system, unexpected situations, or the course of the disease. See the vignette of Sophia (4) for an example how a strong wish to die at home was undermined by sudden complications and absence of pivotal care professionals.

Such scenarios were discussed by interviewees in the prospective group as well. Not having access to professional care when it was most needed, not being listened to, and uncontrollable pain were mentioned by interviewees in relation to complicated deathbeds of others in their (extended) social network, or in relation to their own recent experiences with the healthcare system (e.g., vignette 6, Cliff). It led some to inform and prepare themselves well in advance, for example by becoming active members of patient organizations or signing euthanasia declarations. As one interviewee stated, "I don't want to say later, like my best friend's wife: 'no one listened to him, no one listened to me. And the end was horrible.'" (R15) For other interviewees, care in the last phase of life met or even exceeded expectations, regardless whether it entailed a "natural" or a euthanasia-death (e.g., vignette 11, Marc).

With regard to decisions about euthanasia, similar dynamics and interactions with professional carers came to the fore. Several interviewees in the retrospective group explained how their ill relative had already made an informed choice for euthanasia at an early stage in the disease process. See for example the vignettes of Katherine (3), Peter (5) and Ryan (7). Euthanasia was spoken about as an affirmation of a life view; a way to control the place, moment and manner of dying; and as a way to protect others or oneself from suffering or burdens. Meanwhile, interviewees also spoke about how a choice for euthanasia could become a "natural" death under close guidance of a GP (vignette 5, Peter) and that thresholds for suffering seemed to be dynamic (vignettes 1, Nicole, and 5, Peter).

Other interviewees, however, explained how euthanasia had been spoken about only in vague terms with family members and professionals, followed by a time of not talking about impending death, and then surfacing again as an important option in the very last phase of life. See for example the vignette of Ashley (9). Furthermore, this tension between talking and not talking about (future) decisions, and between focusing on the now versus the future, was spoken about in several interviews, and it could relate to both patients and relatives. See for example the vignettes of Cliff (6) and Marc (11). The vignette of Marc also provides an interesting example of the memories that words such as "palliative" may evoke and how this may impact talking about decisions.

Last, when a euthanasia-request was granted, the care-context could still interfere with hopes and plans. Some euthanasia-deaths occurred, due to the specific course of events, with less time for saying goodbye than hoped for, or with more ambivalent feelings than expected. See for example the vignettes of Ryan (7) and Christina (8). Some interviewees also gave examples how "little" mistakes made in the transition from hospital to home led to painful situations for patients and relatives, such as heavy vomiting by a patient just before a euthanasia-death because of a nasal catheter that had mistakenly been removed by the hospital staff before the patient was transferred to his home.

### Navigating strong emotional experiences, before and after (assisted) death

The third element that stood out was the strong emotional experiences that the relatives needed to navigate: those of themselves and of their ill relative, both before and after (an assisted) death. The "emotional work" of family members related to assisted dying specifically has been described before [7]. However, from our narrative interviews it emerged how the "emotional work" of relatives also encompasses dealing with the illness of their loved ones itself, the many changes it involves, and the memories from the (far) past it invokes. See for example the vignette of Cliff (6). In addition, it became clear how these strong emotional experiences, especially when not properly recognized or addressed, could have considerable impact on all decision-making dynamics and the aftermath of any death, whether it entails euthanasia or not

Furthermore, euthanasia is often associated with a planned and controlled farewell. Previous research has nevertheless already shown varied emotional responses to an assisted death [7,8]. The interviews in our study spoke to this unpredictability in a specific way. We came across several interviewees who already had had experience with euthanasia and

were rationally fully supportive, while their emotional response had been more ambivalent. See for example the vignette of Ryan (7), who told how his father's euthanasia death had been "tough, eventhough I had witnessed euthanasia before".

Several interviewees had to literally find their way through overwhelming emotions during the interview-encounter itself, see for example the vignette of Charlotte (9) Others spoke about the emotional struggles they had encountered over time, for example when learning about the diagnosis of incurable cancer or when sudden complications occurred (vignettes 1, Nicole, and 5, Peter). Profound physical and mental changes in their ill relatives as a result of the disease also invoked many emotions. "That his personality changed, too, I found especially hard to bear." (R9) This was often expressed with strong images, such as "sawed off at the ankles" (R15), or "as if the ground beneath me disappeared" (R17). Another interviewee spoke about seeing his wife transform from a "gym junkie" to "somebody lying in bed. (...) It's difficult to witness, but even more difficult to accept" (R16). Some other interviewees noticed the great differences among their siblings in their emotional reaction to the diagnosis or impending death of a parent. "I just cried, my brother walked off" (R12).

Several interviewees mentioned how the illness of their spouse had triggered strong emotions related to previous traumatic events—previous deathbeds or other major life-events—and how this impacted in turn their talking about current affairs (vignette 6, Cliff). In different interviews it came to the fore as well how attending healthcare professionals seemed unaware of relatives' previous traumatic experiences. Some interviewees whose ill relative had died through euthanasia also mentioned how previous traumatic events seemed to impact their ill relative's capacity or willingness to speak openly about what they were going through. See for example the vignette of Christopher (2).

Meanwhile, some interviewees seemed to navigate strong emotional experiences relatively well all by themselves, possibly due to their own professional background or self-referral to a professional. See for example the vignettes of Sophia (4) and Peter (5). Interviewees also spoke about positive emotional experiences and gratitude related to shared moments of joy, connectedness or the making of new memories (e.g., special ceremonies, holidays) that could happen despite of, or sometimes even because of, the illness and impending death (vignettes 10, Charlotte, and 11, Marc).

Among the bereaved interviewees, a broad range of emotions again emerged. On the one hand, interviewees expressed thankfulness about peaceful deathbeds, which could be both a euthanasia-death or a "natural" death, and relief that the suffering for one's loved-one was over. See for example the vignettes of Katherine (3), Peter (5), Ashley (9) and Marc (11). On the other hand, interviewees mentioned deep sadness; loneliness; difficulties of coming to terms with unexpected events that had happened in the last phase of life or the suddenness and speediness of a deathbed that left a great emotional strain. See for example the vignettes of Nicole (1), Christopher (2) and Christina (8). As described above, fully anticipated and supported euthanasia-deaths could also leave relatives with ambivalent feelings, which comes to the fore in Ryan's (7) and Peter's vignette (5).

## Navigating intimate relationships: the "I-you-we"

The final insight we gained in this study was that interviewees again and again shifted between enormous concern for their ill loved one, as well as for themselves, and for their shared and familial relationships: we called this *navigating the "I-you-we"*.

Earlier empirical research showed how patients may pursue or forego euthanasia out of consideration for their loved ones. In addition, relatives have been shown to be willing to take up many tasks and responsibilities, and if necessary set aside their own values, to help patients attain an assisted death [7,8]. However, the interactions between relatives and patients seemed to run even deeper in our narrative interviews. The relatives had to navigate their most intimate relationships in many different ways due to illness and impending death of their loved ones. These navigations encompassed questions of (not) knowing, loving, caring for, protecting, (un)burdening the other and oneself. And all choices and events – e.g., acceptance or refusal of homecare, euthanasia, or the mere act of talking about the end-of-life – could interfere with these navigations. See the vignette of Christina (8) for an illustrative example.

Many interviewees spoke about how their intimate relationship with the ill person was still a source of meaning, strength, mutual support, and how shared positive goals were pursued. They also spoke about how they wanted to protect or unburden their ill loved ones and how they had realized that their ill loved ones had been doing the same. In addition, interviewees in both the prospective and retrospective group spoke about the difficulty of being a bystander; of trying to be supportive while at the same time not overtaking matters; and of putting their relative's needs in the center while acknowledging their own needs as well. See the vignette of Marc (11) for an illustration of several of these elements. Some interviewees specifically mentioned the challenge and relational impact of preferring a wait-and-see stance while their ill loved one explored concrete options related to the end of life (vignette 6, Cliff), or exactly the opposite.

In the retrospective group, several interviewees mentioned they had been willing to shoulder many care burdens and were happy to do so, but that additional help from others was often needed. See the vignettes of Nicole (1), Christina (no. 8) and Ashley (no. 9). In addition, (not) pursuing euthanasia could be part of efforts to care for, protect, or affirm one another, as can be seen in the vignettes of Ashley (9) and Marc (11). Some interviewees also acknowledged the varying degrees of closeness that could exist in relation to the dying person, and how that affected how much one understood of the ill person's suffering and considerations about euthanasia. See for example the vignette of Ryan (no. 7).

Furthermore, several interviewees mentioned how they had experienced difficulty in navigating the "I-you-we" in situations when healthcare professionals only focused on the patient and seemed not to acknowledge the position of relatives. For example, a spouse described the difficulties her family experienced at home due to her husband's mental changes (i.e., irritability, anger), how her husband held off all help, and how the hospital's staff kept referring to his choices alone when she tried to address her concerns and needs. "But we had to deal with the consequences. (...) It would have helped me enormously if someone had said: I understand you. Instead, it was always 'your husband is sick, you need to take this into consideration.'" (R9)

Last, several of the interviews with bereaved spouses revealed how after the death of their partner, a new kind of navigating the I-you-we seemed to begin. Several interviewees mentioned how the bereaved did not want to burden their children with their loneliness or grief (e.g., vignettes 2, Christopher, and 8, Christina). Others, however, found solace in intimate relationships with good friends, siblings or a new partner.

### Navigating with: who accompanies the relatives?

We previously suggested that practices of euthanasia should be considered not in the patient-physician dyad, but in a patient-physician-family triad [8]. However, the narrative interviews invited us to further broaden our perspective. The interviews gave insight into the great significance of the patient-family dyad for all end-of-life navigations including euthanasia. In addition, the interviews showed how there can be many others than physicians –both lay people and professionals— who accompany relatives and patients throughout their navigations when (euthanasia) death is near. And these people seem important when we consider both the problems and possibilities that relatives and patients may encounter during their navigations. Furthermore, we came to see how "the" physician or "the" GP in Dutch end-of-life practices including euthanasia may not exist. GPs are thought to have long-lasting relationships with patients and their close relatives. They are also thought to provide a great deal of continuity. Still, several interviews showed how this may not be the case due to very practical reasons: holidays, retirements, shortages of GPs, spouses with different GPs, or families themselves who search out for a new GP due to issues with the former one.

First and foremost, the interviewees and their ill relatives accompanied *each other* through the different dimensions of life and decisions to be made, including euthanasia. Second, *other relatives and friends, including deceased ones,* accompanied the interviewees: both in a literal, practical and metaphorical sense, and in more supportive and non-supportive ways. Most vignettes show examples of this "accompaniment" by deceased others: memories to the death-beds of beloved others often serve as an important frame of reference, provide meaning to the current situation, or are an important source of strong emotions. See for example the vignettes of Nicole (1), Christopher (2) and Cliff (6).

Several interviewees spoke about *medically trained family and friends* who were of great support in times when they could not get access to care deemed immediately necessary (e.g., vignette 11, Marc). The *broader social network (including patient organizations) and a supportive work environment* also appeared as an important source of emotional and practical support before and after (assisted) death; this was further highlighted by the stories of interviewees who missed this kind of support (e.g., vignette 8, Christina).

With regard to healthcare professionals, different things came to the fore. While the research originally focused on the situation at home, interviewees spoke about important conversations and interactions with *hospital, nursing home and hospice staff (both physicians, nurses, as others)* that influenced further thinking about and trust in decisions and care at the end of life, including euthanasia. See for example the affirmation by medical specialists in the vignette of Christopher (2).

Some interviews illustrated the *possibly enormous valuable role of one well-known GP* in organizing comprehensive care for both relatives and patients when (assisted) death is near. This is illustrated in the vignettes of Peter (5) and Marc (11). Noteworthy is that in these cases, there were no sudden interruptions of the GP's care on crucial moments. In addition, these interviewees already seemed to have to many resources to navigate successfully through strong emotions, hospitals or workplaces respectively, which seems at hand in the vignette of Katherine as well.

Other interviews highlighted the *vulnerability of the key-position of GPs.* In the vignettes of Nicole (1) and Christopher (2), the attending GP changed late in the course of the disease and the decision-making dynamics. While most of the interviewees spoke positively about these "new" GPs, their role seemed to be somewhat more marginal. Other vignettes, such as the ones of Christina (8) and Ashley (9), raise the question whether both the "new" and "old" GPs are able to recognize and act upon difficulties that relatives may encounter while navigating strong emotions and intimate relationships.

This uncertainty about what to reasonably expect from "the" GP in the care for both the patient and relatives themselves was a major topic of concern in the prospective interviews, with some telling examples in the vignettes of Cliff (6), Sophia (4) and Charlotte (10). In the latter vignette, the mere availability of GPs in a certain area also arose as an issue. Another interviewee described their GP as "the missing person" (R18) in the search for support on considerations regarding the end of life. In addition, one interviewee with both a parent and parent-in-law with metastasized cancer described the differences between the two attending GPs she interacted with: from being very proactive to very reactive in the care for the ill parents and their spouses.

In several interviews, the valuable support (both practical, medical-technical and psychological) of *specialized palliative care teams* and *homecare or night care nurses* was mentioned, also in the case of a euthanasia-death (vignette 9, Ashley). Meanwhile, these caregivers could also be experienced as intrusive, and some interviewees mentioned how this kind of care was refused from the start. Some interviewees spoke about the added value of *psychologists*, but often added that they had to seek this care themselves. Others mentioned the lack of counseling on mental aspects both for themselves and their ill relatives. One interviewee honestly admitted that he would likely not have accepted the help of a *pastoral counsellor* if it had been offered to him, but that he ultimately found great emotional support from such a professional when he met one by chance at the workplace.

Furthermore, several interviewees mentioned the valuable contribution of *funeral undertakers* who helped them cope with practical and emotional challenges around the date of (euthanasia) death of their relative, for example in the case of Christina (8). Finally, some interviewees expressed that they had been or are *actively searching for support*, but were not sure where to find it, what to expect from it, or when and how to get access to it (e.g., because of waiting lists). And thereby—paradoxically, because it was certainly not intended as such—BR in her role as *researcher* became a form of care liaison for these relatives, by responding to questions and referring to publicly available resources and other professionals.

## Discussion

The research presented in this article started out as an exploration of "family involvement" in the Dutch practice of euthanasia, in the broader context of end-of-life decision-making and care at home under guidance of GPs. However, through the examination

of narrative interviews with cancer patients' relatives, our perspective on the topic started to shift. It is not relatives who suddenly "get involved in" GP-led euthanasia decision-making and performance. Instead, it is the relatives who have to navigate many different dimensions of life – different spaces, decisions dynamics and unfolding of professional care, strong emotional experiences, intimate relationships – when the death of a loved one is near. In the midst of this, euthanasia can become one of several pressing questions and medical possibilities to relate to. Seen from this perspective, it is medical professionals, including GPs, who get involved in family matters that transcend the individual patient, and that may continue long after his/her death.

Our study shows how relatives may navigate the challenges involved in the impending death of a loved one relatively well, regardless of whether their loved one dies through euthanasia or a "natural" death. They may have their own resources and find sufficient support from their own social network, a GP and other professionals. However, our study shows this is not a given. It depicts the decision-dynamics and care as sometimes more context—or provider-dependent, or less family—oriented than described in earlier qualitative research that addressed only euthanasia [16,24]. Relatives may need recognition of their needs and responsibilities that span beyond the medical domain. Or they may need more concrete support on the emotional and relational level, or more information about available care-provisions and the up- and downsides of it in real-life settings. And if there is a problem with the accessibility, continuity or quality of GPs' care, both patient and relatives may suffer from the consequences, especially due to the GPs' pivotal role in the organization of care at the end of life.

For the Dutch, an urgent question for further research is which experiences as highlighted in our study are currently the norm. What is the quality and continuity of the "medical involvement" and support for relatives in general, regardless of the mode of dying? Due to the qualitative nature of this study and the small sample size, generalizations cannot be made. Nevertheless, wider concerns exist about the mere availability of GPs and the impact of it on end-of-life care [28,29]. Relatives' reduced emotional wellbeing, care-burdens and unaddressed needs have received more attention lately as well [30–32]. Our study indicates that it might be important to take social determinants and regional differences into account in further research: some relatives seem to have more opportunities to successfully navigate the different dimensions of life and to find the needed support than others.

Although our study focused specifically on the Dutch situation, the findings may support further research and reflection on assisted dying end-of-life care in other legislative settings as well. In line with research on assisted dying from Canada, our study highlights how merely implementing assisted dying legislation and organizing comprehensive care for all involved seem to be two different things [33–34]. In addition, our study provides examples of the mixed and context-dependent bereavement experiences of relatives after assisted dying, in line with a recent study from Quebec on grief after medical aid in dying versus "natural death with palliative care" [35].

Furthermore, the examples in our study about "navigating the I-you-we" and the place of assisted dying therein, may provide further empirical insight in the complex relationship between feelings of being a burden and wishes to die [36–37]. Noticeable in this regard are the examples we found of the combination refusal of professional homecare, high care burden on relatives, and a choice for euthanasia. These elements deserve further attention of those with practical and theoretical interest in assisted dying.

As mentioned before, the in-depth study of relatives' narratives made us shift our perspective: from questioning "family involvement" in euthanasia as primarily a medical affair, to critically assessing "medical involvement" in families' navigations through many dimensions of life when the (assisted) death of a loved one is near. This shift in perspective resembles recent pleas of other scholars to view death and dying as profoundly social-relational experiences, instead of merely individual and medical matters [38–40]. Our narrative interview study enriches these conversations by providing nuanced empirical data from a country with almost 30 years' experience in legalized assisted dying.

It is important to note that we do not intend to contest the legitimacy of healthcare professionals' perspectives. On the contrary, we give their perspectives due consideration in the substudies of GPs and other professionals that we have published separately. However, we do question whether the professionals' perspective may have been too prominent in earlier research and theoretical reflections, including our own.

A comprehensive discussion of what the proposed shift in perspective entails for further moral theorizing about assisted dying goes beyond the scope of this article. Nevertheless, we can already offer some directions. Earlier moral concerns related to euthanasia and relatives remain valid, yet seem nevertheless incomplete. With regard to "undue influence" of relatives on patient's decisions, we could recommend to explore how both patients and relatives can be sufficiently supported throughout all dimensions of life when death is near, so that the (emotional) pressure on all of them is reduced and well-balanced decisions can be reached. In addition, we may reflect on the extent of care-responsibilities towards relatives, especially if one sees the relatives of today as the possible patients of tomorrow [38].

Furthermore, we may reflect on whether it is feasible and desirable to make individual physicians such as GPs responsible for counseling both the patient and relatives; for providing both PAD and palliative care; and for overseeing both legal and emotional-relational matters. In other words, this means reconsidering what the "assistance" in "assisted dying" could and should look like, and not just in theory but in everyday practice.

In addition, we may come to a slightly different understanding of moral concerns related to "being a burden" or a "duty to die" and their possible interaction with legalized assisted dying [12,13,41]. We hypothesize that pursuing assisted dying, once legalized and integrated into the healthcare system, can become just one of multiple means for patients and relatives to protect themselves (navigating the I-you-we) or to realize a way or place of dying they desire (navigating spaces; navigating decision dynamics and unfolding of professional care). It may even become a means to deal with strong emotions related to past and/or present experiences. It could have major practical and moral consequences however, if assisted dying would become the most accessible, most assuring, or most accepted way in a society to do so.

Last, the metaphor of navigating we brought up in our study may invoke in some readers the question "navigating whereto?". However, the relatives' navigations could already be seen as an aim in themselves. Relatives try to navigate the different dimensions of life and impending death of their loved one as well as possible despite inevitable adversities and often rapidly changing (care) circumstances. They try to do so with utmost concern for the overall wellbeing of their loved one, while at the same time remaining mindful of the fact that their life will continue after their loved one's death. Interestingly, a separate study of narrative interviews among patients reveal a similar great concern for the wellbeing of their loved ones, as much as for their own lives and losses that continue to unfold (*in progress*). Such a view on the "whereto" invokes the question whether care-systems are flexible and accessible enough to adequately support patients and relatives throughout their highly particular and messy navigations up until and beyond (assisted) death.

Following the epistemology of Walker [21], further exchange among public and professionals is needed to come to more definitive shared understandings about what matters most for patients and their relatives when (assisted) death is near. And we hope that our research can make a valuable contribution to such exchanges in various legislative settings.

## Limitations

While our research-sample was varied with regard to educational level, it did not include stories from relatives with more varied ethnic backgrounds; nor of people who had a more distant relationship with the patient (e.g., cousins, friends) or of relatives of patients in an older age cohort (80–90). More variety in outcomes could thus have been possible. In addition, our study focused on cancer patients' relatives only and the highly specific legislative and care-setting of the Netherlands. We hope that further research can determine if and to what extent our findings are transferrable to other patient groups. Relatives of patients with other illnesses, for instance neurological (ALS, dementia), cardiological, or psychological in nature may have even less predictable trajectories than the ones our respondents described. Furthermore, the mental faculties of these patient could be affected, which further complicates navigating the different dimensions we outline in our results. Families' navigating strong emotions in psychiatric illness and 'voltooid leven' (tired of living) cases may have specific qualities as compared to euthanasia in patients who are terminally ill. Nevertheless, the thick description provided in the vignettes and triangulation with other empirical sources facilitates the transfer of insights to other settings and populations.

Furthermore, we are aware of possible influences of sampling-procedures on study-outcomes: interviewees recruited via patient-organizations might by more critical than average about their healthcare professionals, and vice versa for the interviewees recruited via GPs. However, for both ways of recruitment, we also saw the opposite happening in our study. In addition, we continuously asked ourselves if counter-narratives for specific situations existed in the data set as a whole, and sought for a balanced presentation of contrasting experiences.

In addition, some may question our approach of not actively asking about euthanasia in the prospective group of participants. A more direct approach could have been used as well, especially given the fact that euthanasia is already widely spoken about in the Dutch public sphere. However, we knew from clinical practice that relatives can be quite ambivalent about wanting to talk about euthanasia and other end-of-life choices. We intended to hold space for this ambivalence in our study and to explore this aspect of 'euthanasia decision-making' as well. And we think this approach yielded interesting insights. Following the results of our study, we could imagine how relatives may be so much occupied with navigating strong emotions, relationships and responsibilities in the now, that it severely affects their possibilities to engage in decision-making about the near future, including euthanasia. Researchers and clinicians active in end-of-life care and/or euthanasia could further build on these insights.

Finally, as in all qualitative research, editing and sequencing of the original interview-data took place for the sake of presentation in the article. We chose the specific form of vignettes to present our results. The creation of vignettes requires some form of restorying: editing parts of the original interview-sequence for analytical and pragmatic reasons such as readability. Constant comparison with the original interviews and the analysis, and discussion with fellow-researchers on the choices made, guided this process. However, one cannot prevent some insights getting "lost in editing." Noteworthy is for example that, while all vignettes display seemingly coherent stories, some interviewees had great difficulty describing their experiences coherently. This could be seen as a result on its own, and also one of relevance for healthcare practitioners who try to attune to relatives' stories.

## Conclusion

This narrative interview study illustrated how relatives of Dutch cancer patients often have much more to consider than just the question whether to support a loved one's choice for euthanasia or not. They have more to lose than "just" a spouse, parent or sibling. Relatives need to navigate various dimensions of life when a loved one's (euthanasia) death is near, including strong emotional experiences and intimate relationships. These confrontationational navigations and negotiations often continue even after a loved one has died. Relatives can find support in their own social networks, GPs and other professionals, but this help is not a given. The results of this study made us shift the perspective from euthanasia as a merely medical and individual affair, to both euthanasia and regular end-of-life care as parts of multilayered social-relational experiences of death and dying. Thereby, different ethical and practical questions come to the fore, for example questions about real-life access to care and support for both patients and relatives regardless of the final mode of dying.

## Supporting information

**S1 File. vignettes.**
(DOCX)

## Acknowledgments

We would like to express our thanks to D. Tange and Stichting Kanker.nl for their support at the start of this study; to all the participants, who were willing to share about their lives; to M. Trappenburg for her feedback during the first stages of data gathering and -analysis; to A. van Gemeren for her advice on matters related to oncology while writing the vignettes; and to several fellow-researchers who read this article for membercheck- and review-purposes and who provided very valuable feedback.

## Author contributions

**Conceptualization:** Bernadette Roest.

**Data curation:** Bernadette Roest.

**Formal analysis:** Carlo Leget, Bernadette Roest, Megan Milota.

**Investigation:** Bernadette Roest, Megan Milota.

**Methodology:** Bernadette Roest, Megan Milota.

**Supervision:** Carlo Leget, Megan Milota.

**Writing – original draft:** Bernadette Roest.

**Writing – review & editing:** Carlo Leget, Bernadette Roest, Megan Milota.

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
