## [Decision Letter · Decision Letter 0]

Dear Dr. Leget,

Thank you for submitting your manuscript to PLOS ONE. After careful consideration, we feel that it has merit but does not fully meet PLOS ONE’s publication criteria as it currently stands. Therefore, we invite you to submit a revised version of the manuscript that addresses the points raised during the review process.

We look forward to receiving your revised manuscript.

Kind regards,

Stefaan Six, Ph.D.

Academic Editor

PLOS ONE

Journal Requirements: When submitting your revision, we need you to address these additional requirements. 1. Please ensure that your manuscript meets PLOS ONE's style requirements, including those for file naming. The PLOS ONE style templates can be found at https://journals.plos.org/plosone/s/file?id=wjVg/PLOSOne_formatting_sample_main_body.pdf and https://journals.plos.org/plosone/s/file?id=ba62/PLOSOne_formatting_sample_title_authors_affiliations.pdf 2. In the online submission form, you indicated that your data is available only on request from a third party. Please note that your Data Availability Statement is currently missing contact details for the third party, such as an email address or a link to where data requests can be made. Please update your statement with the missing information.  3. Please include captions for your Supporting Information files at the end of your manuscript, and update any in-text citations to match accordingly. Please see our Supporting Information guidelines for more information: http://journals.plos.org/plosone/s/supporting-information.

Reviewers' comments:

**Comments to the Author**

1. Is the manuscript technically sound, and do the data support the conclusions?

Reviewer #1: Yes

Reviewer #2: Yes

2. Has the statistical analysis been performed appropriately and rigorously?

Reviewer #1: Yes

Reviewer #2: N/A

3. Have the authors made all data underlying the findings in their manuscript fully available?

Reviewer #1: Yes

Reviewer #2: Yes

4. Is the manuscript presented in an intelligible fashion and written in standard English?

Reviewer #1: Yes

Reviewer #2: Yes

Reviewer #1: I really loved reading this manuscript about the family involvement in the Dutch practice of euthanasia.

I think it should be mandatory reading for all healthcare professionals who are involved in end-of-life care.

However, I must confess that I also had ambiguous feelings about the article while reading it. One moment, I thought it was too long, too much information, and I felt it could be more concise. However, I myself could not really think of what should be deleted. At the same time, I regularly felt that a reference to a vignette should be replaced by more quotes from the interviewee.

The conclusion is, I guess, that it reads more like an interesting book chapter than as a scientific article - which I mean in a very positive way.

Other positive aspects:

- good subtitels/subsections/themes

- great that you sought representing relatives' voices in healthcare etc

- good discussion of limitations

The very minimal suggestions/questions I still have are:

1. Occasionally there are two spaces or lack of a full stop or a space between the last word and the full stop, but I suppose the final editing will solve this

2. Why is there such a big variation in interview duration (27-98min)

3. How did you get the interviewees to talk about euthanasia if BR did not actively ask about euthanasia if the participants did not explicitly mention it?

4. Some sentences that seem to be incorrect:

- BR did not actively ask about euthanasia IT during the interview if participants did not first explicitly mention it themselves because of research-ethical considerations

- With regard to “undue influence” of relatives on patient’s decisions, we could as how both patients and relatives can be sufficiently supported throughout all dimensions of life when death is near

- Why did gender seem to be such an important parameter ?

- Vignette 6: what is GP surgery?

- I feel like the conclusion might still be a bit stronger, with perhaps a firmer recommendation towards the health care professionals: they can play a big role in the end-of-life care, but they are only one part of the whole process that a patient and their relatives are confronted with. The difficult situation for the relatives does not end after the patient has died. And I really loved how the article described the situation as being 'multilayered', that term could also be used in the conclusion. (just a suggestion)

Reviewer #2: Thank you very much for the opportunity to review this interesting manuscript. It deals with an important topic and is characterized by an appealing methodological approach. In order to emphasize these aspects even more clearly, I would like to suggest the following changes:

General

- The study focuses solely on cancer patients, but the rationale for this selection remains unclear. It should be explained already in the background section together with the characteristics of this patient group, e.g. in comparison to neurological patients. Their significance for the transferability of the results to other disease groups should be discussed later.

Introduction

Page 3: This narrative interview study showed us that…The results of the narrative interview study invited us to shift our perspective: … we may consider the perspective that it is GPs and other professionals who get involved in death and dying as a family affair

- These are results of the study, which I would see in the corresponding section, rather than in the introduction.

Methods:

- The explanation of the epistemological basis is very well formulated.

Page 3: The research question that guided this sub-study was: what can we learn from co-constructed interview narratives about the needs and felt responsibilities of cancer patients’ relatives during euthanasia decision-making in the broader context of end-of-life care in the home-setting? In particular, we were interested in how relatives’ needs and responsibilities take shape in interaction with their GP.

- This text should be located at the end of the introduction.

Page 3: The study was conducted as part of a larger research-project (50 interviews in total) exploring the needs and responsibilities of patients, GPs and other professionals working in primary care

- Did this study focus on euthanasia? Please specify.

In The Background you write: … we decided to further explore family involvement in the Dutch practice of euthanasia in the broader context of end-of-life care at home under guidance of GPs, by performing a narrative interview study.

And then in the Methods section: From the information leaflet and conversation with BR, the participants knew that euthanasia could be one of the topics in the interview. However, in the prospective group, BR did not actively ask about euthanasia it during the interview if participants did not first explicitly mention it themselves because of research-ethical considerations, even though euthanasia is widely spoken about in the public sphere (see subsection Dutch context and definitions).

- This procedure and its impact on the results should be discussed methodically.

Results

Page 8: … it was remarkable to see how many interviewees had to deal with acute and chronic health issues of their own while caring for their ill relatives, …

- I would replace “many” with ‘some’ or something similar, as many is very quantifying in a qualitative study. Same for “frequently and spontaneously referred to navigating workplaces…” � Better use e.g. “several“.

- It is not transparent if statements come from the prospective or the retrospective interviewee group. Why? This procedure is not justified or discussed.

- It also becomes not clear, how the experiences of euthanasia and regular assisted dying are identical or if there are possibly differences.

- The results section is very long: in my opinion, the descriptions of the main topics could benefit from cutbacks. What are the key messages? Is it possible to illustrate them with tables or figures?

- It is sometimes difficult to understand how the statements relate to the data. I would suggest adding all vignettes to the manuscript as an appendix and only inserting short summaries, indirect or direct quotes in the text where they contribute to understanding.

- The results are already contextualised on the basis of the literature in this section. What is the new content that the discussion brings to this? A stronger separation of the presentation and discussion of results would be helpful.

Discussion

Page 16: See Figure 1 and 2 for a visual representation of this change in perspective.

- I don't understand the figures and what (additional) information they should provide.

- Subheadings could help the reader to follow your arguments.

- Clarify the scope (cancer patients) and discuss special features compared to other groups.

**Do you want your identity to be public for this peer review?** For information about this choice, including consent withdrawal, please see our Privacy Policy

Reviewer #1: **Yes: ** Jan Bollen

Reviewer #2: No

---

## [Author Response · Author response to Decision Letter 1]

25 Mar 2025

We uploaded a response to reviewers among the files.

---

## [Decision Letter · Decision Letter 1]

Navigating life when a loved one’s (euthanasia) death is near: a narrative interview study from the Netherlands

PONE-D-24-39050R1

Dear Dr. Leget,

We’re pleased to inform you that your manuscript has been judged scientifically suitable for publication and will be formally accepted for publication once it meets all outstanding technical requirements.

Kind regards,

Stefaan Six, Ph.D.

Academic Editor

PLOS ONE

Additional Editor Comments (optional):

Although reviewer 5 provides some suggestions for modifications to the manuscript, I am confident that the manuscript is ready to accepted, and leave it up to the authors whether to implement any of the reviewers' proposals.

Reviewers' comments:

Reviewer's Responses to Questions

**Comments to the Author**

Reviewer #3: All comments have been addressed

Reviewer #4: All comments have been addressed

Reviewer #5: (No Response)

2. Is the manuscript technically sound, and do the data support the conclusions?

Reviewer #3: Yes

Reviewer #4: Yes

Reviewer #5: Partly

3. Has the statistical analysis been performed appropriately and rigorously?

Reviewer #3: N/A

Reviewer #4: N/A

Reviewer #5: N/A

4. Have the authors made all data underlying the findings in their manuscript fully available?

Reviewer #3: Yes

Reviewer #4: No

Reviewer #5: No

5. Is the manuscript presented in an intelligible fashion and written in standard English?

Reviewer #3: Yes

Reviewer #4: Yes

Reviewer #5: Yes

Reviewer #3: The editor's and reviewers' suggestions were rigorously followed. As a result, the article gained strength. One minor suggestion:

The moral epistemology of MU Walker formed the foundation of the empirical study described in this article (Walker, 2007). => The moral epistemology of Walker (2007) formed the foundation of the empirical study described in this article.

Reviewer #4: (No Response)

Reviewer #5: The main merit of the paper is to consider the family of a patient who requests (or considers) euthanasia, not merely as an "influencing factor", but to investigate the emergence and dynamics of death wishes in the context of the familiy’s life as such. However, if one considers the respective length of the text sections of the paper, the focus shifts to examining the various aspects of the challenges of accompanying a dying person by the family as a general topic. In a country where euthanasia has been practised for a very long time, it may be understandable to see the issue of euthanasia as one of many other challenges that the family of a dying person is confronted with. However, the description of the various aspects of “navigation” in the results section is very detailed and the frequent reference to the vignettes (which were not available to the reviewer) makes this section difficult to read. This reviewer is of the opinion that significant parts of the results section could be shortened without losing the essential message. Since the “conclusions” take up the topic of euthanasia again as the main message, it would perhaps be possible to also focus the “results” and “discussion” more in this direction.

Monor point: the authors write that the study is part of a larger research project (p.3). Were all of these interviews (50) conducted in the manner described? How were the interviews analysed here selected from this pool?

**Do you want your identity to be public for this peer review?** For information about this choice, including consent withdrawal, please see our Privacy Policy

Reviewer #3: **Yes: ** Dr Dennis Demedts

Reviewer #4: No

Reviewer #5: No

---

## [Editor Report · Acceptance letter]

PONE-D-24-39050R1

PLOS ONE

Dear Dr. Leget,

I'm pleased to inform you that your manuscript has been deemed suitable for publication in PLOS ONE. Congratulations! Your manuscript is now being handed over to our production team.

Kind regards,

on behalf of

Dr. Stefaan Six

Academic Editor

PLOS ONE